

# Correlation analysis of serum endothelial cell specific molecule-1, endothelial microparticles, hypoxia inducible factor-1α levels and acute myocardial infarction and their predictive value for major adverse cardiovascular events: a retrospective study

Qiubing Zhang[*], Zhe Xu[*], Ping Shi, Jia Zeng, Xiaohong Yin and Fang Gou

Department of Cardiology, Guangyuan Central Hospital, Guangyuan, Sichuan, China
[*] These authors contributed equally to this work.

## ABSTRACT

**Objective**. This study aimed to analyse the correlation of endothelial cell specific molecule-1 (ESM-1), endothelial microparticles (EMPs) and hypoxia inducible factor-1α (HIF-1α) serum levels with the occurrence of acute myocardial infarction (AMI) and determine their short-term predictive value for major adverse cardiovascular events (MACE) following AMI treatment.

**Methods**. Retrospective data analysis was performed on the medical records of 106 patients with AMI admitted to our hospital between October 2020 and October 2022. The control group consisted of 106 healthy volunteers that received a physical examination at our hospital's physical examination centre within the same time frame. ESM-1, EMP and HIF-1α serum levels were compared between the two groups. Independent risk variables for AMI were examined. Furthermore, these individuals were separated into the poor prognosis group ($n = 41$) and good prognosis group ($n = 65$) according to the presence or absence of MACE. Finally, the ESM-1, EMPs and HIF-1α serum levels were correlated with the development of MACE in patients with AMI, and the predictive value of serum ESM-1, EMPs and HIF-1α for MACE was evaluated. The serum HIF-1α, EMP and ESM-1 levels were significantly different between the two groups ($P < 0.05$). Multivariate logistic regression analysis revealed the elevated serum levels of HIF-1α (odds ratio (OR) = 1.819), EMPs (OR = 1.071) and ESM-1 (OR = 14.655) as AMI risk variables. A substantially significant ($P < 0.05$) correlation was found between the elevated levels of serum HIF-1α (OR = 18.716), EMPs (OR = 26.185) and ESM-1 (OR = 13.992) and the development of MACE in patients with AMI. According to receiver operating characteristic (ROC) curve analysis, the combined evaluation value of HIF-1α, EMPs and ESM-1 serum levels in predicting MACE was high with an area under the curve (AUC) of 0.931.

**Conclusion**. Patients with AMI have abnormally high ESM-1, EMP and HIF-1α levels in their serum, all of which have been linked to the development of MACE. Together, these parameters have high sensitivity and specificity for early MACE identification.

Corresponding author
Fang Gou, gfzqb413@163.com

## INTRODUCTION

Acute myocardial infarction (AMI), the ischaemic necrosis of the myocardium, is an acute and critical type of coronary heart disease. Its clinical manifestations are persistent severe retrosternal pain, often accompanied by a sense of impending death and fever. Cardiac troponin I (cTnI) is the key biomarker of AMI (*Kala et al., 2024*). Major adverse cardiovascular events (MACEs) are a substantially hazard to the mental and physical well-being of patients with AMI and may occur after therapy. MACEs include recurrent myocardial infarction, cardiac failure, angioplasty and mortality. Therefore, their early diagnosis has important clinical significance. Hypoxia inducible factor-1α (HIF-1α) (*Dölling et al., 2022*), endothelial microparticle (EMP) (*Yuan, Maitusong & Muyesai, 2020*) and endothelial cell specific molecule-1 (ESM-1) (*Wei et al., 2022*) levels are associated with AMI. HIF-1α is sensitive to hypoxic environments; its level rapidly increases when the body's myocardial cells are in a state of ischaemia and then decrease after correction. EMPs are mainly synthesised by endothelial cells in hypoxia or inflammatory environment and can further accelerate the atherosclerotic response of the body. ESM-1 is mainly secreted by activated endothelial cells and closely related to cardiovascular diseases, and its expression is regulated by various cytokines. This study aimed to examine the association of ESM-1, EMP and HIF-1α serum levels with AMI and their predictive value for MACEs in 106 patients with AMI admitted to our hospital between October 2020 and October 2022. The subsequent outcomes are given as follows.

## DATA AND METHODS

### General data

The medical records of 106 patients with AMI admitted to our hospital from October 2020 to October 2022 were retrospectively analysed. The criteria for inclusion were as follows: ① from 2020 through October 2022, patients who met the AMI diagnosis standards established by the European Society of Cardiology and the American College of Cardiology (*Alpert et al., 2000*) and confirmed by electrocardiogram and high-sensitivity troponin examination for the unified definition published for AMI, increase and/or decrease in myocardial necrosis markers (preferred cTn) and an increase in at least one of the values by more than the reference normal upper limit 99 percentile, accompanied by at least one of the following types of evidence: symptoms of myocardial ischaemia; electrocardiogram indicating new ischaemic changes, such as new ST-T changes or new left bundle branch block, electrocardiogram indicating the formation of new pathological Q waves and imaging evidence suggesting new surviving myocardial loss or new wall motion abnormalities; ② first onset and ③ complete medical records available for analysis. The exclusion criteria were as follows: ① cardiac insufficiency, ② congenital heart disease, ③ old myocardial infarction, ④ complicated with malignant tumour and ⑤ combined with

systemic immune diseases. In the AMI group, 106 patients (48 men and 58 women, with a mean age of $63.32 \pm 6.98$ years) fulfilled the selection requirements; among them, 55 were diagnosed with hypertension. Our hospital's physical assessment department recruited healthy participants to serve as the study's control group. The inclusion criteria were as follows: ① normal electrocardiogram examination and ② complete medical records and follow-up data available for analysis. The exclusion criteria were as follows: ① cardiac insufficiency, ② congenital heart disease, ③ old myocardial infarction, ④ complicated with malignant tumour and ⑤ combined with systemic immune diseases. A total of 106 healthy volunteers (49 men and 57 women) with an average age of $63.28 \pm 6.88$ years fulfilled the requirements for the control group; among them, 49 were diagnosed with hypertension. No statistically significant variation ($P > 0.05$) in hypertension ratio, gender or age was found between the two groups. This study was conducted in accordance with the ethical regulations of the Declaration of Helsinki. Guangyuan Central Hospital agree to ethical investigation (No. ZXYY2020-08-013), and the hospital's ethics committee agreed to waive informed consent.

## Methods
### Follow-up
All the patients who suffered AMI had their follow-up data collected through the hospital's computer system. The follow-up was mainly conducted by the outpatient department or *via* telephone once every 2 weeks for 6 months, and the occurrence of MACEs in patients during the follow-up was recorded. MACEs included heart failure, cardiogenic shock, arrhythmia and death. According to the presence or absence of MACEs, the patients were separated into the good prognosis group ($n = 65$) and poor prognosis group ($n = 41$).

### Detection of blood routine parameters and hs-CTnI, serum HIF-1α, EMP and ESM-1 levels
The blood routine parameters and serum HIF-1α, EMP and ESM-1 levels of patients with AMI on the second day after admission were collected by the information collection system of the hospital. The blood routine parameters included red blood cell (RBC), haemoglobin (Hb) and platelet (PLT) levels, measured using a fully automated haematology analyser (model BC6800; Shenzhen Mindray Bio-Medical Electronics Co., Ltd., Shenzhen, China). Hs-CTnI levels were measured using the Getein1600 fluorescence immunoassay analyser (Getein Biotech, Inc., Nanjing, China). Serum EMP, HIF-1α and ESM-1 levels were detected by CyFlow Cube8 flow cytometer (Shanghai Huanxi Medical Instrument Co., Ltd., Shanghai, China), chemiluminescence immunoassay and ELISA (Shanghai Hengfei Biotechnology Co., Ltd., Shanghai, China), article number: EK0752), respectively.

## Statistical analysis
Statistical power analysis was conducted using G*Power version 3.1.9.7. In accordance with Cohen's guidelines, a medium effect size ($d = 0.5$) and 0.05 alpha error rate (*i.e.*, significance level) were set. The sample sizes for the two groups were 65 and 41, respectively. With the use of this software, the actual statistical power achieved under these conditions was 0.801. This result indicated that our study design has sufficient sensitivity to detect the

**Table 1 Serum HIF-1α, EMP and ESM-1 levels of the two groups (X ± S).**

| Clinical data | Acute myocardial infarction (AMI) group ($n = 106$) | Control group ($n = 106$) | Statistic | P |
|---|---|---|---|---|
| HIF-1α (ng/L) | 44.54 ± 8.98 | 21.25 ± 5.37 | 22.915 | 0.000 |
| EMPs (/μL) | 379.65 ± 55.16 | 207.32 ± 43.13 | 25.337 | 0.000 |
| ESM-1 (ng/mL) | 1.19 ± 0.37 | 1.01 ± 0.05 | 5.046 | 0.000 |

**Table 2 Logistic regression analysis of serum HIF-1α, EMPs, ESM-1 and AMI.**

| Hazard | β | Standard error (SE) | Ward χ2 | P | Odds ratio (OR) (95% confidence interval (CI)) |
|---|---|---|---|---|---|
| HIF-1α | 0.598 | 0.120 | 4.973 | 0.000 | 1.819 (1.500–2.428) |
| EMPs | 0.069 | 0.012 | 5.516 | 0.000 | 1.071 (1.050–1.104) |
| ESM-1 | 2.685 | 0.638 | 4.206 | 0.000 | 14.655 (4.508–55.906) |

hypothesised effect size while maintaining the probability of a type I error within an acceptable range.

The data were analysed using SPSS21. For a comparison of the two groups, independent sample $t$-test was performed on the data that followed a normal distribution (indicated by x ± s). The number of cases or the rate was used to symbolise the quantitative data, and the $\chi 2$ test was applied to compare the means of the two groups. Meanwhile, the Kruskal–Wallis rank sum test was employed to compare the means of more than two. Relevant determining variables of AMI were analysed using a multivariate logistic regression model, and the diagnostic value was assessed with a receiver operating characteristic (ROC) graph. The cutoff for statistical significance was set at $P < 0.05$.

## RESULTS

### Comparison of serum HIF-1α, EMP and ESM-1 levels between the AMI group and control group

The hs-CTnI levels in the AMI group were 232.64 ± 67.22 ng/L. The differences in serum HIF-1α, EMP and ESM-1 levels between the two groups were statistically significant ($P < 0.05$) (Table 1).

### Logistic regression analysis of serum HIF-1α, EMPs, ESM-1 and AMI

Multivariate logistic regression analysis was performed with the incidence of AMI as the dependent variable (0 = No, 1 = Yes) and the factors showing statistical importance in the binary analysis as the independent variables. Table 2 displays that the elevated serum levels of HIF-1α (odds ratio (OR) = 1.819), EMPs (OR = 1.071) and ESM-1 (OR = 14.655) were all independent risk factors for AMI ($P < 0.001$).

**Table 3  Examines the correlation between general information and MACEs in individuals with AMI.**

| Clinical data | Good prognosis (*n* = 65) | Poor prognosis (*n* = 41) | Statistic | *P* |
|---|---|---|---|---|
| Sexuality (n (%)) | | | 0.520 | 0.471 |
| Male | 30 (46.15%) | 16 (39.02%) | | |
| Female | 35 (53.85%) | 25 (60.98%) | | |
| Age (X ± S, years) | 63.18 ± 8.21 | 63.12 ± 8.17 | 0.031 | 0.975 |
| Body mass index (BMI) (X ± S, kg/m$^2$) | 25.87 ± 2.67 | 25.76 ± 2.54 | 0.209 | 0.835 |
| Number of diseased vessels (n (%)) | | | 2.546 | 0.280 |
| Single branch | 26 (40%) | 20 (48.78%) | | |
| Double branch | 17 (26.15%) | 13 (31.71%) | | |
| Three branches | 22 (33.85%) | 8 (19.51%) | | |
| Hypertension (n (%)) | 31 (47.69%) | 24 (58.54%) | 1.184 | 0.276 |
| Diabetes (n (%)) | 28 (43.08%) | 22 (53.66%) | 1.130 | 0.288 |
| Smoking (n (%)) | 30 (46.15%) | 20 (48.78%) | 0.070 | 0.792 |

**Table 4  Treatment details in individuals with AMI (n (%)).**

| Clinical data | Good prognosis (*n* = 65) | Poor prognosis (*n* = 41) | Statistic | *P* |
|---|---|---|---|---|
| Percutaneous coronary intervention (n (%)) | 56 (86.15%) | 36 (87.8%) | 0.060 | 0.807 |
| Beta-blocker (n (%)) | 55 (84.62%) | 33 (80.49%) | 0.304 | 0.581 |
| Renin-angiotensin system (RAS) blocker (n (%)) | 55 (84.62%) | 34 (82.93%) | 0.053 | 0.818 |

## Analysis of the relationship between general information and MACE in patients with AMI

No substantial difference ($P > 0.05$) in demographic characteristics such as gender, age, body mass index, prevalence of vascular disease, hypertension, diabetes or smoking situation was found between the two groups (Table 3). Additionally, no significant differences in treatment details and blood routine parameters were observed between the two groups ($P > 0.05$) (Table 4 and Fig. 1).

## Analysis of the relationship of serum HIF-1α, EMP and ESM-1 levels with MACE in patients with AMI

A considerably significant difference ($P < 0.05$) was found between the serum HIF-1α, EMP, and ESM-1 levels and the incidence of MACE in the patients with AMI (Table 5). These findings suggest that elevated levels of HIF-1α, EMPs, and ESM-1 are associated with a poor prognosis in patients with AMI, indicating their potential as biomarkers for predicting MACEs.

## Predictive value of serum HIF-1α, EMP and ESM-1 expression for MACE

The ROC curve was generated using the serum amounts of HIF-1α, EMPs, and ESM-1 in individuals with AMI as the independent variable and MACE as the dependent variable (1 = Yes, 0 = No). A high diagnostic value was found for the combined evaluation of HIF-1α,

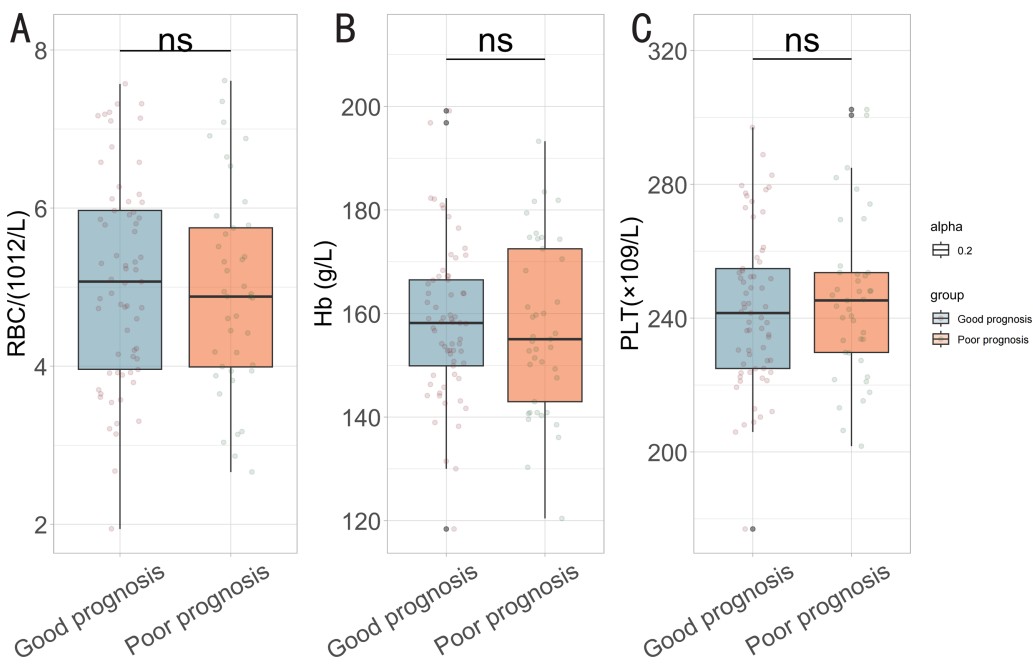

**Figure 1  Comparison of blood routine parameters between the two groups.** (A) Red blood cell (RBC); (B) haemoglobin (Hb); (C) platelet (PLT). ns: no statistically significant difference.

**Table 5  Relationship of serum HIF-1α, EMPs and ESM-1 with MACE in patients with AMI (X ± S).**

| Clinical data | Good prognosis (n = 65) | Poor prognosis (n = 41) | Statistic | P |
|---|---|---|---|---|
| HIF-1α (ng/L) | 31.65 ± 8.54 | 47.65 ± 10.32 | 8.662 | 0.000 |
| EMPs (/μL) | 298.65 ± 50.32 | 397.65 ± 57.87 | 9.305 | 0.000 |
| ESM-1 (ng/mL) | 1.06 ± 0.19 | 1.35 ± 0.47 | 3.760 | 0.000 |

**Table 6  Predictive value of serum HIF-1α, EMPs and ESM-1 expression for MACEs.**

| Index | AUC | 95% CI | Optimal cut-off value | Specificity | Sensitivity |
|---|---|---|---|---|---|
| HIF-1α | 0.784 | 0.820–0.927 | 44.615/(ng/L) | 0.680 | 0.852 |
| EMPs | 0.896 | 0.954–0.979 | 390.000/(/μL) | 0.720 | 0.938 |
| ESM-1 | 0.746 | 0.005–0.275 | 1.245/(ng/mL) | 0.720 | 0.802 |

EMP, and ESM-1 serum levels in predicting MACE (area under the curve (AUC) = 0.931) (Table 6 and Fig. 2).

## Logistic regression analysis of serum HIF-1α, EMPs, ESM-1 and MACE

Multivariate logistic regression analysis was performed with the incidence of MACE as the dependent variable (0 = No, 1 = Yes) and the factors showing statistical importance in the binary analysis as the independent variables. Table 7 displays the results of the logistic regression analysis using ROC cutoff points, indicating that the elevated serum

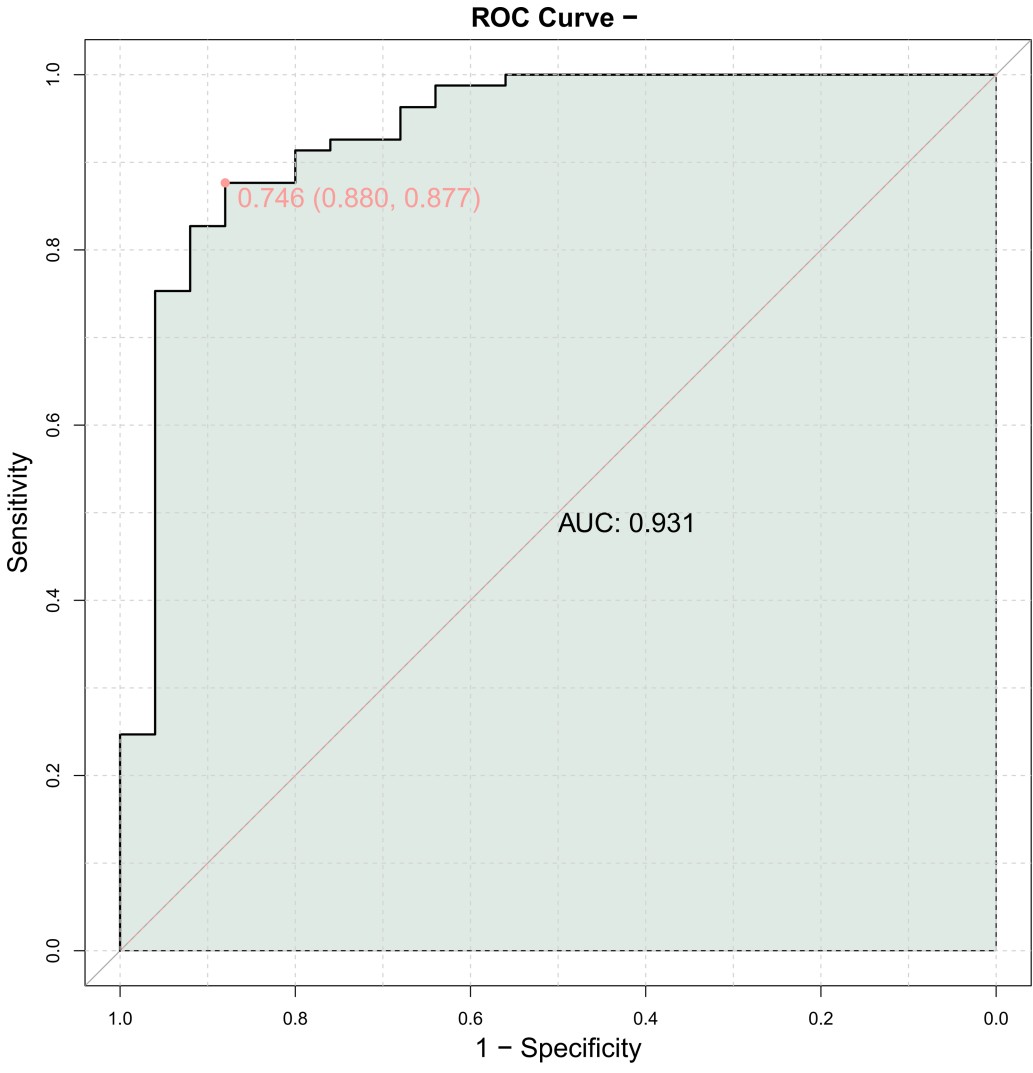

**Figure 2** Predictive value of serum hypoxia inducible factor-1α (HIF-1α), endothelial microparticle (EMP) and endothelial cell specific molecule-1 (ESM-1) expression for major adverse cardiovascular events (MACE).

levels of HIF-1α (OR = 18.716), EMPs (OR = 26.185) and ESM-1 (OR = 13.992) were all independent risk factors for MACE ($P < 0.001$).

## DISCUSSION

AMI is a common clinical cardiovascular disease mainly caused by acute coronary artery occlusion for various reasons, resulting in partial coronary artery stenosis and myocardial tissue ischaemia, hypoxic necrosis or softening. If standard treatment is not given timely, this condition may lead to arrhythmia, heart failure and other serious heart diseases which can greatly affect the life of patients (*Ducrocq et al., 2021*; *Damluji et al., 2021*). Therefore,

**Table 7  Logistic regression analysis of serum HIF-1α, EMPs, ESM-1 and MACE.**

| Hazard | β | SE | Wardχ2 | P | OR (95% CI) |
|--------|------|------|-------|------|-------------|
| HIF-1α | 2.929 | 0.755 | 3.879 | 0.000 | 18.716 (4.260–82.224) |
| EMPs | 3.265 | 0.770 | 4.241 | 0.000 | 26.185 (5.789–118.435) |
| ESM-1 | 2.638 | 0.899 | 2.934 | 0.003 | 13.992 (2.401–81.528) |

the early diagnosis of AMI and MACE can provide basis for clinical implementation of timely and effective prevention and treatment measures.

Logistic regression analysis revealed that high serum HIF-1α (OR = 1.819), EMP (OR = 1.071), and ESM-1 (OR = 14.655) levels were all found to be independent risk variables for AMI. A high diagnostic value was indicated by the ROC curve, which showed that the combined measurement of blood HIF-1α, EMP, and ESM-1 levels for predicting MACEs had an AUC of 0.931. MACEs are strongly linked to serum HIF-1α (OR = 18.716), EMPs (OR = 26.185) and ESM-1 (OR = 13.992).

Microparticles (MPs), also known as microvesicles, are lipid membrane vesicles (0.1–1 μm in diameter) released by cells (including platelets, white blood cells, endothelial cells, red blood cells and vascular smooth muscle cells) during activation or apoptosis (*Luo et al., 2017*). When released, MPs obtain the surface antigen and cytoplasmic components of the mother cells, which enable them to mark the changes of the mother cells and serve as a marker for various diseases. Moreover, MPs can facilitate information transmission between cells by carrying mRNA and microRNA, thus playing an important biological role and even participate in the occurrence of diseases (*Ma et al., 2023*). EMPs are mainly produced due to endothelial cell dysfunction. In addition to being a substitute for early endothelial cell dysfunction, EMPs are an important biological molecule regulating inflammation and coagulation after early endothelial cell injury (*Mörtberg et al., 2019*). AMI causes a series of inflammatory reactions in the body which can stimulate the mass accumulation of calcium ions and activate calpain, thereby causing the cleavage and recombination of endothelial cytoskeleton in the anaerobic environment; EMPs are the products of the recombination of endothelial cells (*Liang et al., 2018*).

The genes implicated in angiogenesis, metabolism, arterial tone control and erythropoiesis can be activated by hypoxia inducible factor-1 (*Feng, Zhan & Ma, 2021*), a transcription factor that is reactively produced when intracellular oxygen partial pressure decreases. HIF-1α and HIF-1β make up a combination of two proteins. They are both members of the family of polypeptides known as basic helix-loop-helix (*Wu et al., 2019*). The 826-amino-acid-long HIF-1α protein acts as a trans-transcriptional regulator of erythropoietin, lactate dehydrogenase, vascular endothelial growth factor and inducible nitric oxide synthase (*Jiang et al., 2018*). In a normal environment, HIF-1α has a low level, exists only in the cytoplasm and has a stable molecular structure which is not easily degraded. Its main function in the body is to promote vascular remodelling and improve collateral circulation. However, HIF-1α is extremely sensitive to hypoxia. After AMI, the hypoxia becomes serious with blood vessel stenosis. At this time, the level of HIF-1α increases rapidly, thus activating a variety of cytokines in the body, improving collateral

circulation ability, vascular remodelling and the state of myocardial ischaemia and hypoxia (*Li et al., 2021*).

ESM-1 is a dermatan sulphate proteoglycan with numerous biological functions, and its synthesis increases in response to neovascularisation or endothelial cell damage. ESM-1 is primarily secreted by endothelial cells (*Wei et al., 2021*). *Dikalova et al. (2020)* showed that when cardiovascular abnormalities occur, the pro-inflammatory factors in vascular endothelial cells induce ESM-1 upregulation through protein kinase C/nuclear factor $\kappa$B signalling pathway.

This study found that the occurrence of MACEs in patients indicated the presence of hypoxia and ischaemia, and the increased levels of HIF-1$\alpha$, EMPs and ESM-1 provided a basis for the clinical prediction of MACEs in patients with AMI. The high expression of HIF-1$\alpha$, EMPs and ESM-1 might be important predisposing factors for the occurrence of MACE in patients with AMI. Our study brings several novel aspects to the literature on AMI and MACE. The combined predictive value of serum levels of ESM-1, EMPs, and HIF-1$\alpha$ for MACE has not been extensively explored previously. The AUC of 0.931 for the combined biomarkers indicated their high diagnostic value, suggesting that this multi-marker approach can improve risk stratification for patients post-AMI. Additionally, the roles of these biomarkers in the pathophysiology of AMI are highlighted, offering insights into their mechanisms and potential as therapeutic targets. This comprehensive analysis contributes to further understanding of the disease and may inform future clinical strategies.

## CONCLUSIONS

Patients with AMI have abnormally high levels of ESM-1, EMPs, and HIF-1$\alpha$ in their serum, all of which have been linked to the development of MACE. Together, these parameters have high sensitivity and specificity for early MACE detection. Given that this study was retrospective and all the patients were from our institution, the possibility of selection bias cannot be ruled out. Additional research is required to validate these findings. Further investigation and confirmation are also warranted because the precise mode of action is still unknown.

### Funding
This study was supported by the Guiding Science and Technology Project of Guangyuan City, Sichuan Province (Grant number: 21ZDYF0047). The funders had no role in study design, data collection and analysis, decision to publish, or preparation of the manuscript.

### Grant Disclosures
The following grant information was disclosed by the authors:
Guiding Science and Technology Project of Guangyuan City, Sichuan Province: 21ZDYF0047.

## Competing Interests

The authors declare there are no competing interests.

## Author Contributions

- Qiubing Zhang conceived and designed the experiments, analyzed the data, authored or reviewed drafts of the article, and approved the final draft.
- Zhe Xu conceived and designed the experiments, analyzed the data, authored or reviewed drafts of the article, and approved the final draft.
- Ping Shi performed the experiments, prepared figures and/or tables, and approved the final draft.
- Jia Zeng performed the experiments, prepared figures and/or tables, and approved the final draft.
- Xiaohong Yin performed the experiments, prepared figures and/or tables, and approved the final draft.
- Fang Gou conceived and designed the experiments, analyzed the data, authored or reviewed drafts of the article, and approved the final draft.

## Human Ethics

The following information was supplied relating to ethical approvals (i.e., approving body and any reference numbers):

This study was conducted in accordance with the ethical regulations of the Declaration of Helsinki. Guangyuan Central Hospital agree to ethical investigation (No. ZXYY2020-08-013).

## Data Availability

The following information was supplied regarding data availability:

ok

## Supplemental Information

Supplemental information for this article can be found online at http://dx.doi.org/10.7717/peerj.19111#supplemental-information.

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
