# Peer review of "Correlation analysis of serum endothelial cell specific molecule-1, endothelial microparticles, hypoxia inducible factor-1α levels and acute myocardial infarction and their predictive value for major adverse cardiovascular events: a retrospective study"

_PeerJ, doi:10.7717/peerj.19111_

## Round 0.1 · original submission · Major Revisions

Dear Doctor Zhang,

Thank you for submitting your manuscript to PeerJ. Please consider the reviewers' comments and concerns, particularly reviewer 1, especially regarding the basic reporting.

Sincerely

·

Basic reporting

The study is of course very interesting, but I think it is very promising and, furthermore, since there are three biomarkers, there could be a bias among them. It seems very bold to consider them to confirm an AMI, knowing that we have ultrasensitive troponin, which is very valuable, validated, reproducible and used worldwide. Another transcendental point that this would have to be validated is to have a cut-off and have a high sensitivity and specificity, considering that this pathology often does not give second chances. What I do think could be a short-term predictor of MACE.

Experimental design

The study design is poor and lacks a basic criterion: if you could count on healthy people to make up the control group and this is within your control, the same number should be used, if that is controlled by the researcher. What I do see as a bit delicate is that many patients with acute myocardial infarction have these biomarkers very low, this was reviewed in their Excel spreadsheet, and that is worrying because if the decision depended on these biomarkers, these patients would not have a heart attack and it turns out that they had three-vessel treatment, patient number 5 of Non-MACE, for example. Now in the MACE group, things change considerably. Therefore, I consider that the study cannot focus on the diagnosis of AMI, but it can be fruitful in the MACE group.

Validity of the findings

I think the study could be redesigned to focus more on a prognostic element of MACE in the medium term as proposed by the researchers, rather than being a tool in the diagnosis of an acute myocardial infarction.

Additional comments

First, I would like to thank the authors for their efforts, since there is always a long road to go from putting together a protocol to obtaining results. I find the relentless search that the entire community has to do to improve ourselves very interesting. I believe that this work can have a great value in prognosis rather than in diagnosis. It is still necessary to define why some patients do not have elevated biomarkers. Just yesterday I treated an asymptomatic patient with angina, mild respiratory distress with bilateral pleural effusion and with a suboccluded anterior descending artery. I believe that ultrasensitive troponin is an excellent biomarker and much must be done to overcome it.

·

Basic reporting

Correlation analysis between serum endothelial cell speciûc molecule-1, endothelial microparticles, and hypoxia inducible factor-1³ levels and acute myocardial infarction
Manuscript
The article ‘Correlation analysis between serum endothelial cell speciûc molecule-1, endothelial microparticles, and hypoxia inducible factor-1³ levels and acute myocardial infarction’
This study aimed to analyze the correlation among endothelial cell specific molecule-1 (ESM-1), endothelial microparticles (EMPs), and hypoxia inducible factor-1a (HIF-1a) serum levels and the occurrence of acute myocardial infarction (AMI).
This is an interesting and remarkable study that aims to analyze the correlation between endothelial cell specific molecule-1 (ESM-1), endothelial microparticles (EMPs), and hypoxia inducible factor-1a (HIF-1a) serum levels and the occurrence of acute myocardial infarction (AMI). I believe it will be a better text after the corrections I suggested.
Comments to the author(s)
1- This study demonstrated that patients with AMI have abnormally high ESM-1, EMPs, and HIF-1a levels in their serum, all of which have been linked to the development of MACE. With this inference, should there be a difference in the treatment approach to these patients? My humble suggestion would be to briefly state your opinions on this matter.
2- What do you think your study brings new to the literature?
3- Do you think your study population is sufficient for the results you obtained?
4- Spelling errors in the article should be corrected and spelling rules should be observed.
5- English of the article should be improved.

Experimental design

Do you think your study population is sufficient for the results you obtained?

Validity of the findings

1- Do you think your study population is sufficient for the results you obtained?
2- Spelling errors in the article should be corrected and spelling rules should be observed.
3- English of the article should be improved.

---

## Round 0.2 · accepted · Accept

Dear Dr. Zang,

Congratulations on your acceptance. The reviewers have accepted the final adjustments to your manuscript.

·

Basic reporting

From the point of view of structure, the work has a good framework, as well as the English version, which has been improved, although a new revision is always necessary. As for biomarkers for MACE, more and more extensive work is obviously needed to confirm them. In addition, it must be remembered that this test is molecular and, like all work of this nature, its translation into clinical medicine remains to be seen. The tables and figures are adequate and reflect what the article wants to show. The references are adequate.

Experimental design

The groups are better balanced, although we know that retrospective studies have better scientific value and inspire randomized studies that are the basis of scientific research. The results are promising although we must see their external validity, their reproducibility and their usefulness in everyday life, which is where one ultimately aims with molecular studies.

Validity of the findings

The results are good, especially in the aggregate. It remains to be seen whether these results are reproducible in long-term follow-up.

Additional comments

No